# Soybean Oil Regulates the Fatty Acid Synthesis II System of *Bacillus amyloliquefaciens* LFB112 by Activating Acetyl-CoA Levels

**DOI:** 10.3390/microorganisms11051164

**Published:** 2023-04-29

**Authors:** Qiang Cheng, Zhongxuan Li, Jing Zhang, Henan Guo, Marhaba Ahmat, Junhao Cheng, Zaheer Abbas, Zhengchang Hua, Junyong Wang, Yucui Tong, Tiantian Yang, Dayong Si, Rijun Zhang

**Affiliations:** 1State Key Laboratory of Animal Nutrition, Laboratory of Feed Biotechnology, College of Animal Science & Technology, China Agricultural University, Beijing 100193, China; 2College of Bioengineering, Sichuan University of Science & Engineering, Zigong 643000, China; 3Xinjiang Laboratory of Special Environmental Microbiology, Institute of Applied Microbiology, Xinjiang Academy of Agricultural Sciences, Urumqi 830091, China

**Keywords:** *Bacillus* LFB112, fatty acid metabolism, type II fatty acid synthesis pathway, unsaturated fatty acids

## Abstract

[Background] *Bacillus* LFB112 is a strain of *Bacillus amyloliquefaciens* screened in our laboratory. Previous studies found that it has a strong ability for fatty acid metabolism and can improve the lipid metabolism of broilers when used as feed additives. [Methods] This study aimed to confirm the fatty acid metabolism of *Bacillus* LFB112. Sterilized soybean oil (SSO) was added to the Beef Peptone Yeast (BPY) medium, and its effect on fatty acid content in the supernatant and bacteria, as well as expression levels of genes related to fatty acid metabolism, were studied. The control group was the original culture medium without oil. [Results] Acetic acid produced by the SSO group of *Bacillus* LFB112 decreased, but the content of unsaturated fatty acids increased. The 1.6% SSO group significantly increased the contents of pyruvate and acetyl-CoA in the pellets. Furthermore, the mRNA levels of enzymes involved in the type II fatty acid synthesis pathway of FabD, FabH, FabG, FabZ, FabI, and FabF were up-regulated. [Conclusions] Soybean oil increased the content of acetyl-CoA in *Bacillus* LFB112, activated its type II fatty acid synthesis pathway, and improved the fatty acid metabolism level of *Bacillus* LFB112. These intriguing results pave the way for further investigations into the intricate interplay between *Bacillus* LFB112 and fatty acid metabolism, with potential applications in animal nutrition and feed additive development.

## 1. Introduction

The use of probiotics as an alternative to antibiotics in livestock has been widely studied, with many studies demonstrating their ability to improve growth performance and meat quality [1,2,3,4,5,6], likely due to the enhancement of the gut ecosystem [7,8]. Bacillus species have been used as probiotics for at least 50 years and are commonly employed as growth promoters and competitive exclusion agents in animal production [9]. This is attributed to their high-temperature resistance [10,11], acid and alkali resistance [12,13], ease of culture, storage stability, and low processing loss [13,14,15]. Amongst these species, *Bacillus subtilis* is a typical representative; its subordinate strain, *Bacillus amyloliquefaciens,* can secrete various protein-active substances. The gryA gene is an important taxonomic basis of *Bacillus amyloliquefaciens* [16,17]. Our laboratory early screened a strain of *Bacillus amyloliquefaciens*, named *Bacillus* LFB112, which was further investigated due to its ability to secrete a variety of bacteriocins with broad-spectrum antibacterial activity [18]. Subsequent animal studies revealed that *Bacillus* LFB112 could improve the growth performance of broilers while also improving the structure of intestinal flora and meat quality; particularly noteworthy was the increased content of unsaturated fatty acids (UFA) in muscle tissue observed after supplementation with *Bacillus* LFB112 [19,20]. This finding has prompted us to explore further the relationship between *Bacillus* LFB112 supplementation and the host metabolism related to unsaturated fatty acids.

Bacillus species are able to modify their fatty acid (FA) patterns to adapt to a wide range of environmental changes [21]. Bacillus strains exhibit a well-defined fatty acid synthesis (FAs) II system that is balanced with an FA degradation pathway and regulated to respond efficiently to the needs of the cell [21,22,23]. This type II FAs pathway is the primary pathway for fatty acid metabolism in bacteria and plant cells [24]. Bacillus can produce a variety of cellular metabolites through the type II FAs pathway, including linear and branched chain fatty acids [25] with different chain lengths [26] and saturations [24]. The type II FAs pathway can be divided into the initial and elongation stages. In the initial stage, malonyl-CoA was synthesized with acid-CoA as the substrate under the action of ACC and other enzymes. Then, Acyl Carrier Protein (ACP) was synthesized under the action of apo-ACP, FabD, and FabH [27]. In the elongation stage, acetoacetyl-ACP can synthesize long-chain ACP under the action of FabG, FabZ, FabA, FabI, and other enzymes [28]. Long-chain ACP can further participate in the carbon chain elongation cycle or synthesize fatty acids under the action of Plsx, Plsy, DES, and other enzymes [29]. Cai et al. [30] conducted whole genome sequencing and metabolic pathway analysis of *Bacillus* LFB112, revealing a high abundance of genes involved in energy production and conversion, as well as the transport and metabolism of lipids, carbohydrates, amino acids, etc. Among its 245 metabolic pathways, the biosynthesis ability of fatty acids was found to be particularly pronounced with the potential to synthesize a variety of fatty acids. This led to speculation that *Bacillus* LFB112 may be able to regulate meat quality in broilers by providing more unsaturated fatty acids through its type II FAs pathway. To further investigate this hypothesis, this experiment investigates the fatty acid metabolism of *Bacillus* LFB112 under nutritional pressure in vitro, with the aim of deepening our understanding of the metabolic pathways of *Bacillus* LFB112 associated with fatty acid metabolism. The outcomes of this research have the potential to provide novel insights into the potential applications of Bacillus in the animal husbandry industry, with valuable implications for practical utilization in animal production.

## 2. Materials and Methods

### 2.1. Preparation and Culture of Seed Solution of Bacillus LFB112

The *Bacillus* LFB112 was deposited in the China General Microbiological Culture Collection Center (CGMCC No. 2996). *Bacillus* LFB112 glycerol bacteria were isolated from a bacteriophage tube using a sterile inoculation loop, scribed onto a solid BPY medium, and incubated at 37 °C for 12 h. A single colony was then picked from the solid medium and inoculated into 10 mL of liquid BPY medium. This process was repeated to obtain a single colony, which was subsequently inoculated into a triangular flask containing 100 mL of BPY medium and incubated in a Constant Temperature Bacterial Incubator (LONGYUE, Shanghai, China) (150 rpm) at 37 °C for 12 h to generate the *Bacillus* LFB112 seed solution. The culture conditions for the *Bacillus* LFB112 were 150 rpm and 37 °C under a constant temperature.

### 2.2. Preparation of Bacterial Pellets

A sample of 40 mL of bacterial solution was centrifuged at 8000 rpm for 10 min at 4 °C in a high-speed refrigerated centrifuge. The supernatant was discarded, and the bacterial pellet was collected. The pellets were resuspended with 10 mL of sterile PBS, centrifuged at 8000 rpm for 5 min, and the supernatant was discarded to collect the bacterial pellet. This process was repeated twice more to collect the bacteria. Subsequently, the bacteria were resuspended with 5 mL of sterile PBS and quickly frozen in liquid nitrogen for 3 min before being thawed in a 37 °C water bath. Finally, they were centrifuged again at 8000 rpm for 10 min, and the supernatant was discarded to collect the bacterial pellet.

### 2.3. Determination of Acetyl-CoA in Bacteria by High-Performance Liquid Chromatography (HPLC)

The quantification of the acetyl-CoA concentrations in the *Bacillus* LFB112 was performed using HPLC, following established protocols with slight modifications, based on methods described by Shurubor et al. [31] and Li et al. [32].

Sample pretreatment: The bacterial pellets were dissolved in 2 mL of 6% perchloric acid solution (containing 2 mmol/L dithiothreitols) and subjected to ultrasonic treatment for 15 min. The sample was then centrifuged at 12,000 rpm for 10 min, and the supernatant was transferred to a pre-cooled 10 mL centrifuge tube. Subsequently, 3 mol/L of a potassium carbonate solution was added dropwise to adjust the pH to 3.0 before being centrifuged again at 12,000 rpm for 10 min. The supernatant solution was then filtered with a 0.22 µm sterile filter prior to HPLC detection.

Chromatographic conditions: Column: Shim-pack GIST C18, 5 µm, 4.6 × 250 (Cat. No. 227-30017-08); the column temperature is 25 °C; the wavelength λ = 254 nm; the injection volume is 20 μL; the mobile phase is 200 mmol/L of phosphoric acid sodium dihydrogen solution (pH = 5.0): acetonitrile = 94:6; the flow rate is 0.5 mL/min; and the running time is 30.01 min.

### 2.4. Detection of Short-Chain Fatty Acids (SCFA) Content in Culture Medium by Gas Chromatography (GC)

The quantification of the SCFA contents in the medium was performed using GC, following established protocols with slight modifications, based on methods described by Tangerman et al. [33].

Sample pretreatment: Following aseptic protocols, 1.5 mL of the sample was taken into a sterile centrifuge tube and centrifuged at 12,000 rpm for 15 min at 4 °C. The supernatant was then transferred to a new centrifuge tube and mixed with formic acid and crotonic acid in a ratio of 89:10:1, respectively. The mixture was vortexed to ensure homogeneity before being filtered through a 0.22 µm sterile filter for subsequent GC detection.

Chromatographic conditions: Instrument: Shimadzu GC 2010 Plus; the temperature of the SPL-1 is 250 °C; the split ratio is 20.0; the column is an Agilent DB-FFAR, 30.0 m × 0.32 mm × 0.25 µm (Cat. No. 123-3232); the temperature rise program of the column temperature box is 100 °C for 2 min, 10 °C/min, increase to 180 °C, and hold for 1 min; the total time is 11.0 min. The temperature of the FID detector is 270 °C.

### 2.5. Detection of Medium-Chain Fatty Acids (MCFA) and Long-Chain Fatty Acids (LCFA) Acid Content in Culture Medium and Bacteria by GC

The total lipids were extracted following the chloroform–methanol procedure of Wei et al. [20].

Sample pretreatment: Weigh about 400 mg of the sample into a glass test tube with a rubber stopper, add 1 mL of C11 internal standard, 1 mL of isooctane, and 4 mL of 10% chloroacetyl solution, then cap the tube stopper. Shake and mix well. Put the test tube in a water bath at 80 °C for 2.5 h, and shake and mix every 15 min. After the water bath, take out the test tube and let it cool to room temperature. Transfer the liquid to a 50 mL centrifuge tube, and rinse the tube with 4 mL of potassium carbonate solution; the washing solution is also transferred to the centrifuge tube; centrifuge at 3000 rpm for 5 min, and filter the supernatant solution with a 0.22 µm sterile filter. Then, carry out the GC detection.

Chromatographic conditions: Instrument: Shimadzu GC 2010 Plus; the temperature of the SPL-1 is 220 °C; the split ratio is 10.0; the column is a SHIMADZU SH-Rt-2560, 100.0 m × 0.25 mm × 0.20 µm (Cat. No. 227-36311-01); the temperature rise program of the column temperature box is 75 °C for 2 min, 10 °C/min, increase to 165 °C, hold for 1.50 min, 1.50 °C/min to 185 °C, keep for 5 min, 1.50 °C/min to 195 °C, and keep for 5 min, 3.20 °C/min, the temperature is raised to 240 °C and kept for 16 min; the temperature is raised to 250 °C at 3.20 °C/min and kept for 7 min; the total time is 83.00 min. The temperature of the FID detector is 270 °C.

### 2.6. Detection of mRNA Levels of Genes in Bacillus LFB112 by qPCR

At the corresponding time point, 5 mL of the bacterial solution was collected in a sterile 15 mL centrifuge tube and centrifuged at 8000 rpm for 20 min at 4 °C to obtain the bacterial precipitate. Subsequently, 1 mL of sterile PBS solution was added to the centrifuge tube, and the bacterial precipitate was resuspended using a pipette gun. The resuspended bacteria were then transferred to a new sterile 2 mL centrifuge tube and re-centrifuged at 12,000 rpm for 5 min at 4 °C. The supernatant was discarded, and the process was repeated once. After aspirating the solution in the centrifuge tube, 100 μL of lysozyme solution (3 mg/mL) was added to resuspend the bacteria by gently blowing with a pipette before bathing in water at 37 °C for 10 min.

Total RNA extraction from bacteria was performed using the TIANGEN RNAprep Pure Cell/Bacteria Kit (TIANGEN BIOTECH (Beijing) Co., Ltd., Beijing, China), followed by reverse transcription of the tRNA to cDNA using the PrimeScript RT Reagent Kit with a gDNA Eraser (Takara Biotechnology (Dalian) Co., Ltd., Dalian, China). Finally, the quantitative real-time PCR assay was performed on the Roche LightCycler96 PCR detection system (Roche, Switzerland), according to optimized PCR protocols, using the TB Green Fast qPCR Mix (Takara Biotechnology (Dalian) Co., Ltd.). The primer pairs were used as listed in Appendix A.

### 2.7. Statistical Analysis

All the data were analyzed using SPSS version 21.0 (SPSS Inc., Chicago, IL, USA). The results were expressed as the mean ± the standard error (SEM), and statistical significance was determined using a one-way analysis of variance (ANOVA). Tukey’s multiple comparison test was used for analysis when significant differences were observed between groups. A *p*-value < 0.05 was considered statistically significant. GraphPad Prism version 7.0.0 for Windows (GraphPad Prism 7 for Windows, version 7.0.0, GraphPad Software, San Diego, CA, USA) was used to create graphical representations of the data.

## 3. Results

### 3.1. The Fatty Acid Profile of Bacillus LFB112 in the Supernatant of BPY Medium

In order to gain insight into the fatty acid secretion of *Bacillus* LFB112, the fatty acid content in the medium was first detected (Figure 1). Nine fatty acids with higher contents (C(FA) ≥ 0.1 mg/mL) were identified in the original BPY medium, including palmitic acid, stearic acid, oleic acid, linoleic acid, myristic acid, linolenic acid, trans-linoleic acid, and arachidic acid, as well as erucic acid. After 12 h of inoculation with *Bacillus* LFB112 in the BPY medium, eight fatty acids with higher contents (C(FA) ≥ 0.1 mg/mL) were detected; these included palmitic acid, stearic acid, oleic acid, acetic acid, cardamom acid, trans-linoleic acid, arachidic acid, and erucic acid. After 24 h of inoculation, only four fatty acids with higher contents (C(FA) ≥ 0.1 mg/mL) remained; these were acetic acid, palmitic acid, stearic acid, and isovaleric acid (Appendix A). It can be seen from the above results that, when cultured in BPY medium, *Bacillus* LFB112 can utilize the MCFA and LCFA, such as palmitic acid and stearic acid, in the medium for its own metabolism, and metabolize them to produce SCFA, such as acetic acid and isovaleric acid.

When *Bacillus* LFB112 was cultured for 24 h, the medium’s SCFA were mainly acetic acid, isobutyric acid, and isovaleric acid (Appendix A). Furthermore, the content of acetic acid was higher than that of isovaleric acid and isobutyric acid. Interestingly, during this period, the fatty acids in the medium gradually decreased with the increasing culture time, suggesting that fatty acids in the medium are also a nutrient source for *Bacillus* LFB112. These results indicate that, under normal physiological conditions, *Bacillus* LFB112 will gradually absorb MCFA and LCFA from the medium and secrete SCFA primarily composed of acetic acid, isovaleric acid, and isobutyric acid.

To further investigate the fatty acids metabolism of *Bacillus* LFB112 under different nutritional pressures, soybean oil was added to the BPY medium. The addition of the soybean oil to the BPY medium resulted in a decrease in the SCFA content at both 12 h and 24 h (*p* < 0.05) (Table 1), indicating an inhibition of SCFA production by *Bacillus* LFB112 under nutritional pressure. Additionally, monounsaturated fatty acids (MUFAs), polyunsaturated fatty acids (PUFAs), as well as saturated fatty acids (SFAs) decreased at 12 h, while the MUFAs and PUFAs increased at 24 h with the SFAs decreasing accordingly (Figure 2).

The total amount of fatty acids in the SSO group decreased from 150.422 mg/g to 18.992 mg/g during 0–12 h, with a corresponding decrease in the proportion of PUFAs from 65.49 to 50.83%, and an increase in the proportion of SFAs from 14.77 to 28.75%. In contrast, during 12–24 h, there was an increase in the total amount of fatty acid from 18.992 mg/g to 21.622 mg/g, accompanied by an increase in the MUFAs proportion from 20.42 to 25.83%, and a return of the PUFAs proportion to 63.38%, as well as a return of the SFAs proportion back down to 10.79% (Figure 3).

The results of this study indicate that *Bacillus* LFB112 can rapidly absorb 87.37% of the fatty acids in the culture medium within 0–12 h, with a higher absorption rate for PUFAs than SFAs. During 12–24 h, *Bacillus* LFB112 was observed to secrete fatty acids, primarily monounsaturated and polyunsaturated varieties.

### 3.2. The Fatty Acid Profile of Bacillus LFB112 Pellets

Analysis of the fatty acid content in the *Bacillus* LFB112 pellets revealed that the content of fatty acids in the SSO group was higher than that in the control group. Palmitic acid was found to be the most abundant fatty acid in the control group pellets, while linoleic acid had the highest content among those present in pellets from the SSO group. Furthermore, a decrease in the total fatty acid content was observed with the increasing incubation time (from 12 h to 24 h) for both groups (Figure 4). Palmitic acid, stearic acid, myristic acid, oleic acid, and linoleic acid were the five major components of the control group, and linoleic acid, oleic acid, palmitic acid, linolenic acid, and stearic acid were the five major components of the SSO group, respectively (Appendix A). In the control group, there was a decrease from 32.652 mg/g to 18.539 mg/g of total fatty acids present over the 12~24 h period, with SFAs being predominant (>80%). Similarly, there was a decrease from 653.234 mg/g to 173.772 mg/g of total fatty acids present in the SSO group pellets over the 12~24 h period, with PUFAs being predominant (>52%) (Appendix A).

The results of this study suggest that *Bacillus* LFB112 rapidly absorbs PUFAs, such as linoleic acid and oleic acid, from the BPY medium supplemented with soybean oil during the logarithmic phase (0–12 h). Subsequently, in the plateau phase, fatty acids are secreted by the bacteria into the culture medium. This is evidenced by comparing the fatty acid profiles in vivo and in the supernatant cultures at 12 h and 24 h (Appendix A), which shows an increase in fatty acid content in the culture medium from 12 h to 24 h, while a decrease is observed in cells over this same period.

These findings indicate that *Bacillus* LFB112 has an affinity for the soybean oil present in the culture medium and begins to absorb it during the logarithmic growth. Additionally, acetic acid production was inhibited (*p* < 0.05) when the soybean oil was added to the BPY medium. To further investigate this phenomenon, experiments were conducted using different concentrations of soybean oil and various inducing agents to assess the acetic acid production by *Bacillus* LFB112.

### 3.3. Different Inducers on the Acetic Acid Production of Bacillus LFB112

The effects of different inducers on the acetic acid production of *Bacillus* LFB112 cultured for 0–24 h were investigated (Figure 5). In the control group, *Bacillus* LFB112 was cultured typically in the BPY medium, and its acetic acid production increased with the culture time from 0 to 6 h, reaching a plateau at 6 h, and continuing to increase after 12 h. When 0.5% starch was added to the BPY medium (starch group), the acetic acid production increased with the incubation time from 0–3 h and reached a plateau at 3 h, which was consistent with that of the control group. After 12 h, the acetic acid production continued to increase until 24 h, when it was higher than that of the control group. In contrast, when 4% soybean oil was supplemented in the BPY medium (SSO group), the acetic acid production by *Bacillus* LFB112 followed a similar pattern as that of the control group but had lower levels during the plateau phase compared to those observed in the control group (*p* < 0.05).

Moreover, Figure 6 shows that there is a negative linear correlation between the soybean oil addition amount and the produced acetic acid content. When the soybean oil addition amount ranged from 0~4.0%, the correlation coefficient R^2^ = 0.647, and its linear equation is C(acetic acid) = 1085.018 − 19080.647 × X; while, for the addition range of 0–1.6%, the correlation coefficient R^2^ = 0.756, and its linear equation is C(acetic acid) = 1206.074 − 47391.705 × X. These results again confirm that the addition of soybean oil to the medium inhibits the production of acetic acid by *Bacillus* LFB112.

### 3.4. Effect of SSO on the Glycolysis Pathway of Bacillus LFB112

*Bacillus* LFB112 primarily utilizes the glycolytic pathway to metabolize glucose into acetate, with intermediates such as pyruvate and acetyl-CoA. Previous findings have indicated a decrease in acetic acid production by *Bacillus* LFB112 in the presence of soybean oil, leading to a hypothesis that soybean oil may inhibit the glycolytic pathway and subsequently reduce acetic acid production. To confirm this hypothesis, the organic acid production in the glycolytic pathway was examined (Appendix A). In comparison to the control group, the addition of 1.6% soybean oil did not result in any significant changes to the levels of malic acid, lactic acid, citric acid, and succinic acid in the supernatant medium (*p* > 0.05), suggesting that the presence of soybean oil does not affect the TCA cycle of *Bacillus* LFB112.

The present study next evaluated the effects of soybean oil on the contents of pyruvic acid (PA) and acetyl-CoA and the mRNA levels of relevant enzymes in bacterial cultures. Results from Figure 7 indicate that adding both 0.4% and 1.6% soybean oil to the culture medium showed an increase in the content of acetyl-CoA (*p* < 0.05) and a decrease in the content of PA (*p* < 0.05) after 12 h of culturing. After 24 h of culturing, the addition of 0.4% and 1.6% soybean oil significantly increased the contents of PA and acetyl-CoA (*p* < 0.05).

RT-PCR technology was also employed to detect the mRNA levels of relevant enzymes in the 24 h bacteria. As depicted in Figure 8, the addition of 1.6% soybean oil to the culture medium resulted in the up-regulation of the mRNA levels of pyruvate kinase, lactate dehydrogenase, and acetate kinase (*p* < 0.05) compared to the control group. These results suggest that soybean oil may modulate bacterial acetyl acid metabolism, possibly by affecting the activity of key enzymes involved in the pyruvate and acetyl-CoA pathways.

### 3.5. Effect of SSO on Type II FAs Pathway of Bacillus LFB112

The findings from the conducted experiment reveal that the inclusion of soybean oil into the culture medium did not impede the glycolytic process of *Bacillus* LFB112. The decrease in acetic acid content in the culture medium following the addition of soybean oil raises a question because of the increased content of acetyl-CoA in the cultured samples at 12 and 24 h. This contradiction suggests that acetyl-CoA in *Bacillus* LFB112 has additional destinations other than the production of acetic acid through metabolism. An analysis of the whole genome data of *Bacillus* LFB112 using the KEGG pathway database (https://www.genome.jp/kegg-bin/show_pathway?bamf01100, accessed on 5 March 2023) demonstrated the involvement of acetyl-CoA and pyruvate in the metabolism of acetyl acid and fatty acids in the bacteria. An important observation was that acetyl-CoA and pyruvate are constituents of the preliminary phase of the fatty acid biosynthesis pathway. Consequently, a new hypothesis was proposed, suggesting that the absorption of soybean oil by *Bacillus* LFB112 in the culture medium alters the distribution pathway of acetyl-CoA, activates the bacterial fatty acid biosynthesis pathway, and reduces acetic acid production. Accordingly, the mRNA levels of related proteases in the fatty acid biosynthesis pathway were determined by qPCR, and the details are shown in Figure 9.

The mRNA levels of enzymes related to the initiation and activation of the fatty acid biosynthesis pathway were found to be up-regulated after 24 h of exposure to soybean oil, as evidenced by the increased expression of *pptT*, *fabD*, and *fabH*. Similarly, enzymes related to the elongation stage of the pathway, including *fabG*, *fabZ*, *fabI*, and *fabF*, were also up-regulated. Additionally, termination stage-related enzyme transcripts, such as *acs*, *faa*, and *acl*, were significantly up-regulated (*p* < 0.05). Moreover, the mRNA level of the fatty acid catabolism-related enzyme, *acat-C,* was up-regulated after 24 h of soybean oil exposure (Appendix A).

These results demonstrate that the addition of sterilized soybean oil to the BPY medium effectively activates fatty acid biosynthesis pathways in *Bacillus* LFB112 bacteria.

## 4. Discussion

The widespread recognition of probiotics as feed additives in animal husbandry has ushered in a new era of research that explores their potential for improving the intestinal microflora of broilers and enhancing their digestive function and immunity [34,35,36]. *Bacillus* LFB112 is a strain of *Bacillus amyloliquefaciens,* screened from herbal reagents in our laboratory [18], which has multiple benefits as an animal feed supplement. Our laboratory’s several animal tests have demonstrated its remarkable ability to improve the meat quality of broiler chickens. Specifically, it has been shown to significantly increase the content of unsaturated fatty acids in muscle tissue, resulting in superior meat quality. A study by Yang et al. [37] has provided experimental verification that *Bacillus subtilis* KT260179 supplementation can promote body growth, improve plasma lipid parameters, and enhance breast meat quality in broilers. Additionally, Tang et al. [4] have reported that dietary supplementation with *B*. *subtilis* can improve the carcass traits and meat quality of broilers through an improvement in the fatty acid profile and amino acid composition. These findings are indeed fascinating, and they raise a fundamental question: is there a link between probiotics and host lipid metabolism? In Wei’s experiment [20], we found that the addition of 10^8^ CFU/kg of *Bacillus* LFB112 to the diet increased the monounsaturated fatty acid content by a staggering 36.95% and polyunsaturated fatty acid content by a remarkable 40.54% in the breast muscles of 42-day-old broilers. In another trial conducted by Marhaba [19], we focused on adding 5 × 10^8^ CFU/kg of *Bacillus* LFB112 to the diet to increase serum TG levels in broilers at 39 days of age. The results of probiotics improving fatty acid composition in broiler muscle tissue have also been reported in other literature. Salma et al. [38] added different concentrations of *Rhodobacter capsulatus* to the diet of broilers and found that *R. capsulatus* significantly increased PUFA, MUFA, and UFA/SFA in the thick muscle of broilers after 6 weeks of feeding. The results are highly promising, suggesting that probiotics may have a profound effect on lipid metabolism in broilers. In conclusion, probiotics, specifically *Bacillus* LFB112, have proven to be highly effective as feed additives in animal husbandry, with remarkable potential for improving meat quality and enhancing host lipid metabolism. These findings have significant implications for the livestock industry, as they provide a natural and cost-effective means of enhancing animal health and productivity.

Popova [39] reviewed the research on the chemical composition of meat and showed that the main traits affected by probiotics in the birds’ diet were protein and fat content. Our animal test data has revealed that the addition of *Bacillus* LFB112 to the diet has a profound impact on the levels of unsaturated fatty acids in the serum, liver, and muscle tissues. The implications of this finding are significant, as it raises the intriguing possibility that *Bacillus* LFB112 may have the capacity to biosynthesize unsaturated fatty acids for use by broilers. This, in turn, could lead to a notable improvement in lipid metabolism in these animals. To delve deeper into this matter, an in vitro culture experiment was conducted to analyze the fatty acid metabolism pattern of *Bacillus* LFB112. The hope is that this data will serve as a foundation for subsequent studies aimed at uncovering the molecular intricacies of this fascinating phenomenon.

In order to investigate the types and contents of fatty acids metabolized by *Bacillus* LFB112, we conducted an analysis at three distinct time points during the culture process at 0 h, 12 h, and 24 h. The initial content of fatty acids in the medium, represented by 0 h, was compared to the fatty acids present during the logarithmic phase, represented by 12 h, and the stable phase, represented by 24 h. These time points allowed us to observe changes in the fatty acid content over time. Upon analyzing the results, we observed a noticeable increase in short-chain fatty acid content and a simultaneous decrease in medium- and long-chain fatty acid content as the culture time increased. This suggested that *Bacillus* LFB112 has the ability to utilize fatty acids in the culture medium for metabolic processes, with short-chain fatty acids being produced. Nakano et al. [40] used nuclear magnetic resonance analysis to identify that acetate is one of the main fermentation products of *B. subtilis*. Ramos et al. [41] expound on acetate formation as a general part of the aerobic and anaerobic *B. subtilis* metabolism. Therefore, the production of acetic acid by *Bacillus* LFB112 is a reasonable result. Interestingly, the metabolism of *Bacillus* LFB112 produced short-chain fatty acids, such as acetic acid, isobutyric acid, and isovaleric acid, which was unexpected based on our previous hypothesis regarding the production of polyunsaturated fatty acids. To further investigate this phenomenon, we imported the whole genome data of *Bacillus* LFB112 into the KEGG database and analyzed its lipid metabolism pathway through the KEGG signaling pathway. This analysis revealed that *Bacillus* LFB112 has a relatively complete fatty acid biosynthesis pathway and the potential to synthesize unsaturated fatty acids. Based on these findings, we speculated that *Bacillus* LFB112 may be able to produce unsaturated fatty acids in response to certain nutritional stress stimuli.

Upon analyzing the feed formulae utilized in the animal tests, we discovered a rather intriguing piece of information: the feed formulae included copious quantities of soybean oil. This then beckons the question, would the addition of soybean oil to the feed formulae stimulate the metabolism of *Bacillus* LFB112, thereby leading to the generation of unsaturated fatty acids? It is worth noting that the capacity to utilize a diverse range of carbon sources, encompassing different fatty acids, represents a pivotal aspect of the *Bacillus subtilis* metabolism [42,43]. In the subsequent phase of our study, we proceeded to devise an experiment by supplementing 4% sterilized soybean oil into the basal medium. We meticulously assessed the content of short-chain fatty acids in the culture broth after a period of 12 h and 24 h of incubation. As a consequence, we ascertained that the addition of soybean oil to the basal medium resulted in a significant reduction in the acetic acid content in the culture broth. Moving forward, we proceeded to scrutinize the content of medium- and long-chain fatty acids in the 12 h and 24 h cultures. The results that ensued were nothing short of remarkable. The addition of soybean oil to the basal medium led to a noteworthy augmentation in the content of monounsaturated and polyunsaturated fatty acids, while significantly diminishing the content of saturated fatty acids. This discovery left us fascinated and prompted us to ponder over the possibility of a correlation between the amplification in unsaturated fatty acid content and the subsequent reduction in acetic acid content.

The bacterial metabolism of acetic acid is an intricate process, driven by a plethora of metabolic reactions. These pathways, which include the EMP, HMP, and other pathways, interact in complex and dynamic ways to convert a range of intermediates to acetic acid [44,45]. One of the key steps in this process involves the conversion of pyruvate, a critical metabolic intermediate, into acetyl-CoA. This conversion is driven by the concerted action of two enzymes, pyruvate dehydrogenase and lactose dehydrogenase [46]. Once acetyl-CoA is formed, it is further processed through a series of additional metabolic reactions, including the action of acetate kinase and phosphotransferase, to ultimately produce acetic acid [47].

The results show that soybean oil inhibits the production of acetic acid by *Bacillus* LFB112, so is this by inhibiting the EMP, HMP pathway? An in-depth analysis of the *Bacillus* LFB112 bacteria revealed elevated levels of pyruvate and acetyl-CoA, alongside an upregulation in the mRNA expression of pyruvate kinase, lactose dehydrogenase, and acetate kinase. The findings imply that soybean oil does not pose a hindrance to the acetate metabolic pathway of *Bacillus* LFB112.

In bacteria, in addition to being an end product of glycolytic metabolism, acetic acid can also be involved in the biosynthesis of fatty acids as a substrate [48,49,50]. Microorganisms possessing elevated acetyl-CoA synthetase levels enable the synthesis of acetic acid to acetyl-CoA catalyzed by ACS [47,51], thereby providing the base substance for the type II FAs pathway. In this study, we used qPCR to confirm the upregulation in the mRNA expression level of ACS. Furthermore, we observed a decrease in acetic acid levels and a concomitant increase in acetyl coenzyme A levels in the soybean oil group, suggesting that acetic acid was used by the *Bacillus* LFB112 to synthesize acetyl coenzyme A. Pyruvate and acetyl-CoA, as intermediates, can be used as substrates in the KEGG pathway to activate the type II FAs pathway in bacteria [30,52]. We also verified the mRNA levels of enzymes related to each step of the type II FAs pathway by qPCR and found that soybean oil activated the type II fatty acid biosynthetic pathway in *Bacillus* LFB112.

In summary, the findings provide corroboration for several conjectures: specifically, *Bacillus* LFB112 instigated the type II FAs pathway through the stimulation of the acetic acid-to-acetyl-CoA transformation in the context of soybean oil nutritional stress, eventually engendering the biosynthesis of unsaturated fatty acids. These findings serve to elucidate the intricacies of FA metabolism by *Bacillus* LFB112 and present novel ideas for the exploitation of microorganisms for applications.

## 5. Conclusions

The inhibitory effects of sterilized soybean oil on acetic acid production in *Bacillus* LFB112 cannot be attributed to the suppression of the glycolysis pathway. Rather, the soybean oil appears to modulate the acetyl-CoA distribution pathway in *Bacillus* LFB112, activating the type II FAs pathways, and resulting in elevated levels of unsaturated fatty acids. These intriguing results shed light on the intricate regulatory mechanisms underlying the interplay between SSO and *Bacillus* LFB112, with potential implications for animal nutrition and feed additive development.

## Figures and Tables

**Figure 1 microorganisms-11-01164-f001:**
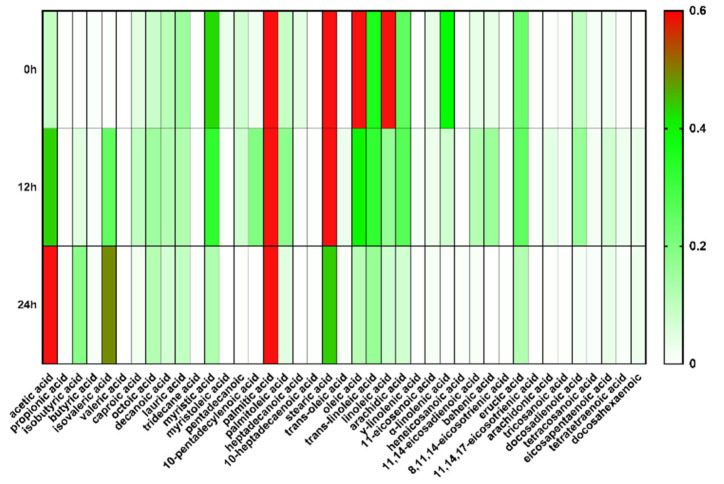
Fatty acid profiles of BPY medium supernatants were determined by GC after 0, 12, and 24 h of incubation in *Bacillus* LFB112. The *Bacillus* LFB112 seed solution of 1 mL was inoculated into 100 mL of BPY culture medium, and the mixture was cultured in a constant temperature bacterial incubator set at 37 °C and 150 rpm. Sampling was conducted at 0, 12, and 24 h of cultivation, and the samples were stored at −20 °C for further analysis. Abbreviations of fatty acids: C16:0, palmitic acid; C18:0, stearic acid; C18:2, cis-9, 12, linoleic acid; C18:1, cis-9, oleic acid; C14:0, myristic acid; C18:3, cis-9,12,15, linolenic acid; C18:2 TT, trans-9,12, trans-linoleic acid; C20:0, arachidic acid; C22:1, cis-13, erucic acid.

**Figure 2 microorganisms-11-01164-f002:**
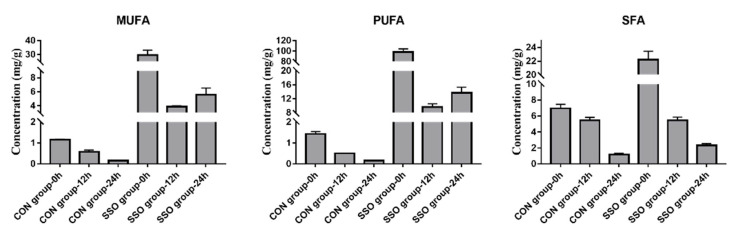
The content of MCFA and LCFA in the supernatant of BPY medium in the control group and soybean oil group medium after 0, 12, and 24 h of culture. Two treatment groups were set up in this experiment: the CON group is the BPY medium without oil + 1.0% *Bacillus* LFB112 seed solution; the SSO group is the BPY medium with 4.0% sterilized soybean oil + 1.0% *Bacillus* LFB112 seed solution. Each group has six replicates. Sampling was conducted at 0, 12, and 24 h of cultivation, and the samples were stored at −20 °C for further analysis.

**Figure 3 microorganisms-11-01164-f003:**
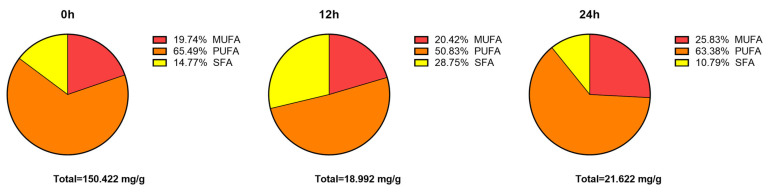
The proportion of fatty acids in the supernatant of BPY medium in the soybean oil group after 0, 12, and 24 h of culture. Two treatment groups were set up in this experiment: the CON group is the BPY medium without oil + 1.0% *Bacillus* LFB112 seed solution; the SSO group is the BPY medium with 4.0% sterilized soybean oil + 1.0% *Bacillus* LFB112 seed solution. Each group has six replicates. Sampling was conducted at 0, 12, and 24 h of cultivation, and the samples were stored at −20 °C for further analysis.

**Figure 4 microorganisms-11-01164-f004:**
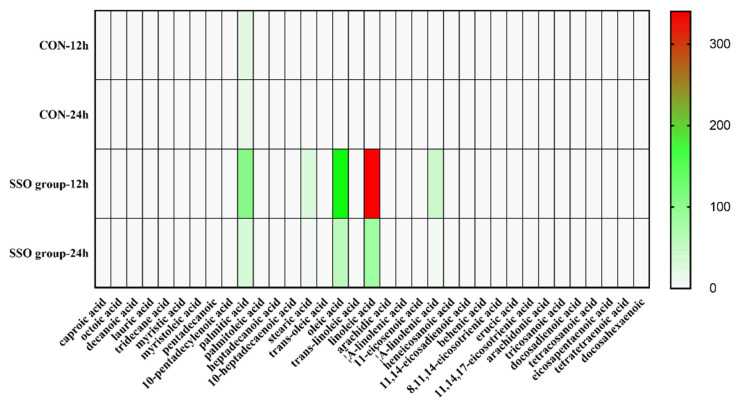
The fatty acids profile in *Bacillus* LFB112 pellets was determined by GC after 12 and 24 h of culture. Two treatment groups were set up in this experiment: the CON group is the BPY medium without oil + 1.0% *Bacillus* LFB112 seed solution; the SSO group is the BPY medium with 4.0% sterilized soybean oil + 1.0% *Bacillus* LFB112 seed solution. Each group has six replicates. Sampling was conducted at 12 and 24 h of cultivation, and the samples were stored at −80 °C for further analysis.

**Figure 5 microorganisms-11-01164-f005:**
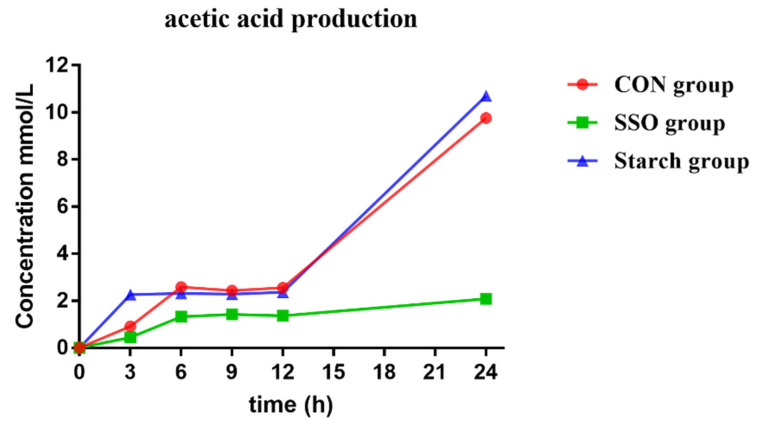
Different inducers on the acetic acid production of *Bacillus* LFB112. In this experiment, three treatment groups were set up, each with six replicates: the treatment groups are the CON group: BPY medium without sterilized soybean oil; and the SSO group: BPY medium with 4.0% sterilized soybean oil; and the starch group: BPY medium with 0.5% starch. Inoculation is performed with 1% seed liquid for all the treatment groups. Samples are collected every 3 h for analysis or observation. The total incubation period is 24 h.

**Figure 6 microorganisms-11-01164-f006:**
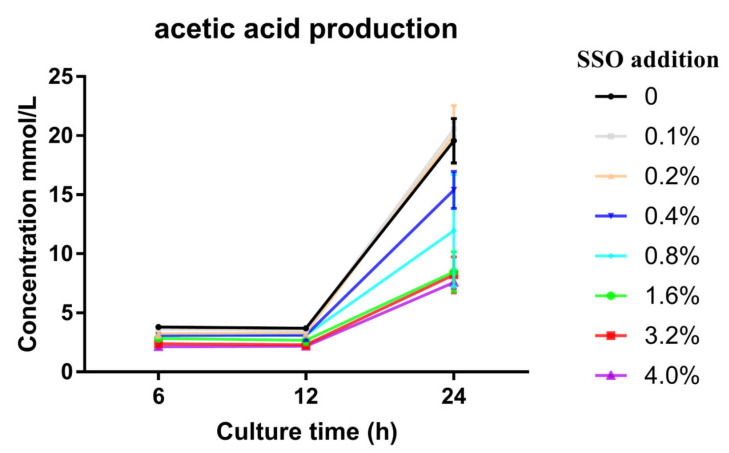
Linear regression analysis between acetic acid production and SSO addition. The experiment involves eight treatment groups with six replicates each. The treatment groups are created by adding different percentages of sterilized soybean oil to BPY culture medium. The percentages of sterilized soybean oil added are 0%, 0.1%, 0.2%, 0.4%, 0.8%, 1.6%, 3.2%, and 4.0%, respectively. Inoculation is performed with 1.0% seed liquid for all the treatment groups, and samples are collected at the 6th, 12th, and 24th hours of cultivation for analysis or observation. linear regression analysis: between 0 and 4.0%, R^2^ = 0.647, C(acetic acid) = 1085.018 − 19080.647 × X; between 0 and 1.6%, R^2^ = 0.756, C(acetic acid) = 1206.074 − 47391.705 × X.

**Figure 7 microorganisms-11-01164-f007:**
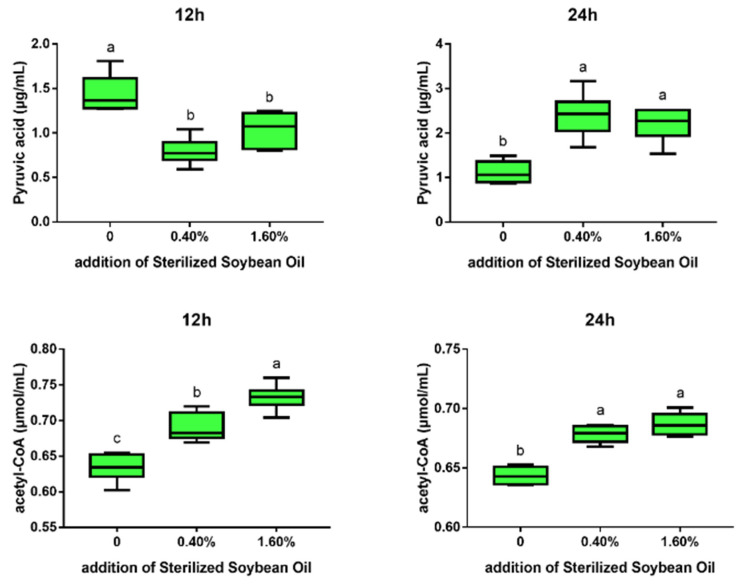
Effect of different concentrations of sterilized soybean oil on pyruvic acid and acetyl-CoA content in bacteria. This experiment set up three treatment groups with six replicates, adding 0%, 0.4%, and 1.6% sterilized soybean oil to the BPY culture medium, respectively, and collected bacterial pellets at the 12th and 24th hours of cultivation. ^a–c^ Different letters on standard error bars indicate a significant difference (*p* < 0.05).

**Figure 8 microorganisms-11-01164-f008:**
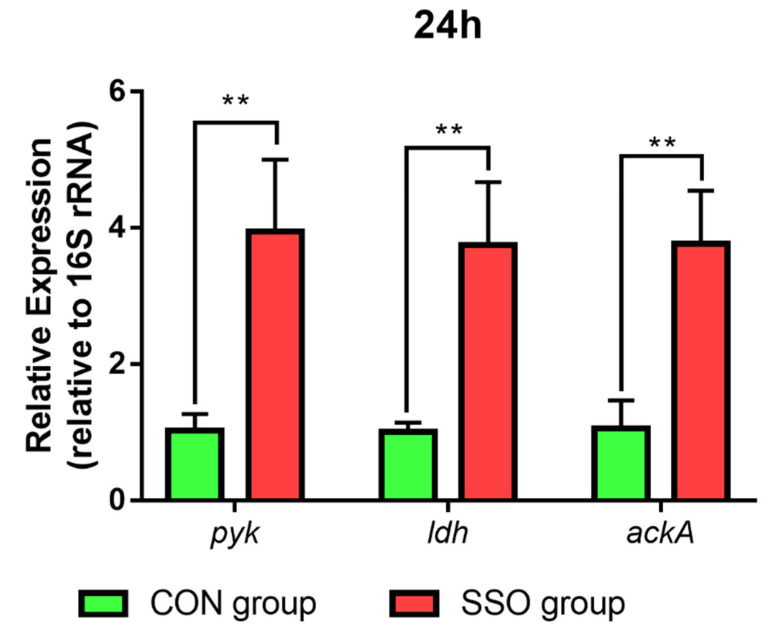
Effect of adding 1.6% soybean oil to the medium on the levels of *pyk*, *ldh,* and *ackA* mRNA in *Bacillus* LFB112 bacteria. This experiment set up two treatment groups with six replicates: the CON group is BPY medium without oil + 1.0% *Bacillus* LFB112 seed solution; the SSO group is BPY medium with 4.0% sterilized soybean oil + 1.0% *Bacillus* LFB112 seed solution. Bacterial pellets were collected at the 12th and 24th hours of cultivation. Abbreviations: *pyk*, pyruvate kinase; *ldh*, lactate dehydrogenase; *ackA*, acetate kinase. The double asterisk (**) symbol is used to indicate statistical significance at the 0.01 level.

**Figure 9 microorganisms-11-01164-f009:**
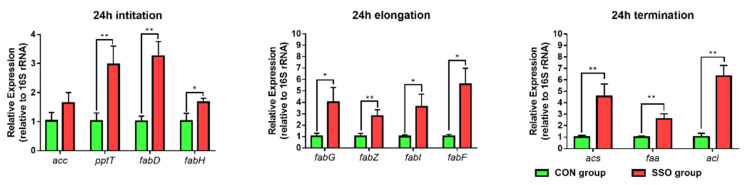
*Bacillus* LFB112 mRNA levels of related enzymes in the FAS II pathway. This experiment set up two treatment groups with six replicates: the CON group is the BPY medium without oil + 1.0% *Bacillus* LFB112 seed solution; the SSO group is the BPY medium with 4.0% sterilized soybean oil + 1.0% *Bacillus* LFB112 seed solution. Bacterial pellets were collected at the 12th and 24th hours of cultivation. Abbreviations: *acc*, acetyl-CoA carboxylase; *pptT*, 4′-phosphopantetheinyl transferase; *fabD*, Malonyl CoA-acyl carrier protein transacylase; *fabH*, 3-oxoacyl-ACP synthase Ⅲ; *fabG*, 3-ketoacyl-ACP reductase; *fabZ*, 3-hydroxyacyl-ACP dehydratase; *fabI*, enoyl-ACP reductase; *fabF*, 3-oxoacyl-ACP synthase II; *acs*, acyl-CoA synthetase; *faa*, long-chain fatty acid-CoA ligase; *acL*, acyl-CoA ligase. The asterisk (*) symbol in the figure is used to indicate statistical significance at the 0.05 level, and the double asterisk (**) symbol is used to indicate statistical significance at the 0.01 level.

**Table 1 microorganisms-11-01164-t001:** The content of SCFA in the supernatant of BPY medium after 12 and 24 h of culture.

Time	Item	CON	SSO	*p*
12 h	Acetic acid	0.261 ± 0.013	0.052 ± 0.008	<0.001
	Isobutyric acid	0.024 ± 0.001	0.002 ± 0.001	<0.001
	Isovaleric acid	0.124 ± 0.006	0.006 ± 0.004	<0.001
24 h	Acetic acid	0.991 ± 0.008	0.185 ± 0.003	<0.001
	Isobutyric acid	0.094 ± 0.004	0.008 ± 0.001	<0.001
	Isovaleric acid	0.382 ± 0.016	0.025 ± 0.001	<0.001

Note: Two treatment groups were set up in this experiment: the CON group is the BPY medium without oil + 1.0% *Bacillus* LFB112 seed solution; the SSO group is the BPY medium with 4.0% sterilized soybean oil + 1.0% *Bacillus* LFB112 seed solution. Each group has six replicates. Sampling was conducted at 12 and 24 h of cultivation, and the samples were stored at −20 °C for further analysis.

## Data Availability

Data sharing is not applicable to this article.

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
