# Peer review of "Soybean Oil Regulates the Fatty Acid Synthesis II System of Bacillus amyloliquefaciens LFB112 by Activating Acetyl-CoA Levels"

_microorganisms, 2023, doi:10.3390/microorganisms11051164_

Round 1

Reviewer 1 Report

Congratulations for your interesting paper.

Some suggestion: 

Line 3 - Bacillus Amyloliquefaciens (Italic, amyloliquefaciens);

Line 31 - change Keywords (different of title);

Line 73 - 75 - aim, maybe put in vitro?

Materials and Methods - it's ok;

Results and Discussion - The figures are presented and discussed, with theoretical reference;

Conclusion - clear and objective

References -  good references.

Author Response

Dear Reviewer,

Thank you for your valuable feedback and comments on my manuscript. We appreciate your thorough review. We have carefully considered your suggestions and have made the following revisions:

Suggestion 1#: Line 3 - Bacillus Amyloliquefaciens (Italic, amyloliquefaciens);

Response: I have referred to the articles in this journal (Title: Discrimination between the Two Closely Related Species of the Operational Group B. amyloliquefaciens Based on Whole-Cell Fatty Acid Profiling) and revised the writing format of bacterial names throughout the manuscript. Thank you for pointing out this oversight, and I will be more attentive to checking the writing format of bacterial names in my future writing.

Suggestion 2#: Line 31 - change Keywords (different of title);

Response: I have carefully considered your comment and revised the keywords as per your suggestion. The revised keywords now better reflect the content of our manuscript and meet your requirements. Keywords: Bacillus LFB112; fatty acid metabolism; Type II fatty acid synthesis pathway; unsaturated fatty acids; Acetyl-CoA.

Suggestion 3#: Line 73 - 75 - aim, maybe put in vitro?

Response: Thank you for pointing out the mistake and providing valuable advice. This experiment is an extension of our previous animal experiments in our laboratory where we found that Bacillus LFB112 promotes the growth of broilers and improves meat quality. The analysis data indicated a relationship between the fatty acid composition in the intestinal tract of broilers and the fatty acid composition in the muscle. To investigate whether the increase in unsaturated fatty acids in the intestinal tract is related to the fatty acid metabolism of Bacillus LFB112, we designed this in vitro experiment. I have modified the purpose of the experiment as follows:

“To further investigate this hypothesis, this experiment investigates the fatty acid metabolism of Bacillus LFB112 under nutritional pressure in vitro, with the aim of deepening our understanding of the metabolic pathways of Bacillus LFB112 associated with fatty acid metabolism. The outcomes of this research have the potential to provide novel insights into the potential applications of Bacillus in the animal husbandry industry, with valuable implications for practical utilization in animal production.”

I believe that this revision clarifies the purpose of the experiment and is more consistent with the in vitro nature of the study. I greatly appreciate your feedback and suggestions, and if you have any further comments or recommendations, please do not hesitate to let me know.

Sincerely,

Cheng Qiang

Reviewer 2 Report

The manuscript entitled (Soybean Oil Regulates the Fatty Acid Synthesis â…¡ System of 2 Bacillus Amyloliquefaciens LFB112 by Activating Acetyl-CoA 3 Levels) is well written but some correct ions needed to be considered for publication.

Some languish and writing editing needed for example:

L 40: ease of culture, storage stability and low processing loss (reference???)

L40: Bacillus subtilis change to Bacillus subtilis

In statistical analysis the authors considered standard deviations instead of standard errors; here the authors must mention the number of samples used in tables or figures fot easy calculation of standard error by readers.

Secondly, it is preferred that not all results expressed as figures, some tables with data of means , number of samples and standard deviations needed.

The materials not contains section for experimental design and groups and this not acceptable, only authors considered analysis and lab methods

The corresponding author makes frequent self-citations along the manuscript and this is not acceptable

Author Response

Dear Reviewer,

Thank you for your valuable feedback and comments on my manuscript. We appreciate your thorough review. We have carefully considered your suggestions and have made the following revisions:

Comments and Suggestions #1: Some languish and writing editing needed for example L 40: ease of culture, storage stability and low processing loss (reference???)

Response: Thank you for pointing out the shortcomings in my manuscript. I have reviewed the sentence and made the following modifications: I added three citations at the end of the sentence to support the statement. Considering that Bacillus can produce spores, which makes it resistant to high temperatures, acids, and alkalis, and thus has the characteristics of easy survival and high storage stability in production applications. These characteristics were also mentioned in the previous citation, so I omitted the annotation when labeling the citation. After receiving your suggestion, I reviewed several literatures and added three citations to support the statement.

Comments and Suggestions #2: Some languish and writing editing needed for example L40: Bacillus subtilis change to Bacillus subtilis

Response: I have referred to the articles in this journal (Title: Discrimination between the Two Closely Related Species of the Operational Group B. amyloliquefaciens Based on Whole-Cell Fatty Acid Profiling) and revised the writing format of bacterial names throughout the manuscript. Thank you for pointing out this oversight, and I will be more attentive to checking the writing format of bacterial names in my future writing.

Comments and Suggestions #3: In statistical analysis the authors considered standard deviations instead of standard errors; here the authors must mention the number of samples used in tables or figures fot easy calculation of standard error by readers.

Response: Thank you for bringing this to my attention. I apologize for the oversight. In my previous studies, I overlooked the meaning of standard deviation and standard error. After consulting various materials, I have come to understand that standard deviation reflects the degree of data dispersion, while standard error measures the reliability of measurement data from the processing group. Using standard error may be more appropriate in analyzing the data, and I will be more mindful of this in my future studies. In the revised version of the manuscript, I have made sure to include the number of samples used in tables or figures. Furthermore, I have corrected the use of standard deviations and replaced them with standard errors in appropriate instances during the statistical analysis.

Comments and Suggestions #4: it is preferred that not all results expressed as figures, some tables with data of means , number of samples and standard deviations needed.

Response: Thank you for your valuable feedback on the data presentation. In the revised version of the manuscript, I will replace the data in Figure 2 with a table presentation, But I am very confused about whether using a table to present other results would be more intuitive. If you have any good suggestions, please let me know and I will change it immediately.

Comments and Suggestions #5: The materials not contains section for experimental design and groups and this not acceptable, only authors considered analysis and lab methods

Response: Thank you for your comments and suggestions regarding the manuscript. I apologize for the oversight in not including a section on experimental design and groups in the materials and methods. Upon reviewing your feedback, I have revised the manuscript to address this issue. In the revised version, I have included a description of the experimental design, including the specific groups used in the study. This information has been added under the relevant chart in the results section. Additionally, I have made sure to reference relevant literature and provide clear explanations of the analysis and laboratory methods used in the study.

Comments and Suggestions #6:The corresponding author makes frequent self-citations along the manuscript and this is not acceptable

Response: Thank you for your criticism of me. I did not take into account the citation rate of the literature. I apologize for the excessive elaboration on the origin and characteristics of the bacteria used in the experiment, which resulted in a high proportion of references. I understand that this may have affected the overall balance and focus of the article. In the revised version of the manuscript, I have taken your comments into consideration and reduced 2 self-references to ensure that the introduction provides relevant and concise background information.

I appreciate your feedback and apologize for any inconvenience caused by the original version. Thank you for your understanding, and I hope that the revised version now better meets the requirements for a well-structured and balanced introduction.

Sincerely,

Cheng Qiang

Reviewer 3 Report

This manuscript can be accepted after a few language check and revisions

Author Response

Dear Reviewer,

Thank you very much for your comments and suggestions on my manuscript, I have made the following modifications to your suggestion.

Comments and Suggestions 1#: English language and style are fine/minor spell check required

Response: Thank you for your feedback regarding the English language and style in my manuscript. I have reviewed the articles in this journal, specifically the one titled "Discrimination between the Two Closely Related Species of the Operational Group B. amyloliquefaciens Based on Whole-Cell Fatty Acid Profiling," and made revisions to the writing format of bacterial names throughout the manuscript accordingly.

I apologize for any oversight in the capitalization of initial letters in the article and appreciate your attention to detail. I have made modifications to ensure that the capitalization is standardized throughout the manuscript.

I will be more vigilant in checking the writing format of bacterial names in my future writing to ensure compliance with the journal's guidelines. Thank you for bringing this to my attention, and I hope that the revised version now meets the required language and style standards.

Sincerely,

Cheng Qiang

Reviewer 4 Report

Manuscript: Soybean Oil Regulates the Fatty Acid Synthesis â…¡ System of Bacillus Amyloliquefaciens LFB112 by Activating Acetyl-CoA Levels. The manuscript is well written. However, a few suggestions below might increase the quality of the manuscript.

Title: Bacillus must be Italics and Amyloliquefaciens: A must be lowercase

Abstract:

The abstract should include more information on the significance of the results, such as how they can be applied to practical applications or contribute to existing knowledge.

The abstract could provide more information on the potential limitations of the study or areas for future research to help readers better understand the scope and relevance of the findings.

Introduction

Line 41: Bacillus must be Italics throughout the manuscript

Objectives are clear

Methods

2.2. Preparation of Bacterial Pellets: provide citation

2.3. Determination of Acetyl-CoA in Bacteria by High-performance liquid chromatography (HPLC): provide citation

2.4. Detection of Short-chain fatty acids (SCFA) content in culture medium and bacteria by gas chromatography (GC): provide citation along with detailed methodology

2.5. Detection of Medium-chain fatty acids (MCFA) and Long-chain fatty acids (LCFA) acid

content in culture medium and bacteria by GC: Provide citation

2.6. Detection of mRNA levels of genes in Bacillus LFB112 by qPCR: provide citation

None of the methodology contains citations.

Results

Line 168: Palmitic acid.. P can be lower-case. Please check this kind of error throughout the manuscript. For example, Line 169 and 172 etc.

Figure 1: quality must be improved, X axis

Discussion

The article primarily cites studies conducted by the authors' laboratory, which may introduce a bias. It would be beneficial to include studies conducted by other research groups to provide a more comprehensive view of the subject.

The article lacks a detailed explanation of the molecular mechanisms underlying the observed effects of probiotics on meat quality and lipid metabolism.

The article lacks a discussion of the potential risks and drawbacks of using probiotics as feed additives. It is important to consider the potential negative effects of probiotics, such as the development of antibiotic resistance and the potential for unintended consequences on the gut microbiota of animals.

Conclusions must be revised to reflect the findings

References are not according to the journal guidelines

It is unclear how the authors confirmed the fatty acid metabolism of Bacillus LFB112 and what specific methods they used to measure the expression levels of genes related to fatty acid metabolism.

Author Response

Dear Reviewer,

Thank you for your valuable feedback and comments on my manuscript. We appreciate your thorough review. We have carefully considered your suggestions and have made the following revisions:

Comments and Suggestions 1#: Title: Bacillus must be Italics and Amyloliquefaciens: A must be lowercase

Response: I appreciate your attention to detail and thank you for bringing this to my attention. I have reviewed the articles in this journal and made revisions to the writing format of bacterial names throughout the manuscript accordingly.

Comments and Suggestions 2#: Abstract: The abstract should include more information on the significance of the results, such as how they can be applied to practical applications or contribute to existing knowledge. The abstract could provide more information on the potential limitations of the study or areas for future research to help readers better understand the scope and relevance of the findings.

Response: I fully agree with your evaluation of my abstract, so I have made modifications to it and re-elaborated on the conclusions and significance of the research.

Comments and Suggestions 3#: Introduction Line 41: Bacillus must be Italics throughout the manuscript

Response: I have made the necessary revisions to ensure that the genus name "Bacillus" is italicized.

Comments and Suggestions 4#: Methods 2.2. Preparation of Bacterial Pellets: provide citation. 2.3. Determination of Acetyl-CoA in Bacteria by High-performance liquid chromatography (HPLC): provide citation. 2.4. Detection of Short-chain fatty acids (SCFA) content in culture medium and bacteria by gas chromatography (GC): provide citation along with detailed methodology. 2.5. Detection of Medium-chain fatty acids (MCFA) and Long-chain fatty acids (LCFA) acid content in culture medium and bacteria by GC: Provide citation. 2.6. Detection of mRNA levels of genes in Bacillus LFB112 by qPCR: provide citation. None of the methodology contains citations.

Response: Thank you for your suggestions on the citation of the experimental method section. I have supplemented the references in the 2.3~2.5 experimental methods section and provided a new description of some experimental operations. I have also added the grouping of the corresponding experiments to the charts of the data results.

Comments and Suggestions 5#: Results Line 168: Palmitic acid. P can be lower-case. Please check this kind of error throughout the manuscript. For example, Line 169 and 172 etc. Figure 1: quality must be improved, X axis

Response: Thank you again for correcting my spelling irregularity, I have revised the entire article to address any discrepancies in capitalization and to change the fatty acid names on the x-axis in Figures 1 and Figures 4 to use the correct fatty acid names instead of the codes.

Comments and Suggestions 6#: Discussion The article primarily cites studies conducted by the authors' laboratory, which may introduce a bias. It would be beneficial to include studies conducted by other research groups to provide a more comprehensive view of the subject.

Response: Thank you for your criticism of me. I did not take into account the citation rate of the literature. I apologize for the excessive elaboration on the origin and characteristics of the bacteria used in the experiment, which resulted in a high proportion of references. I understand that this may have affected the overall balance and focus of the article. In the revised version of the manuscript, I have taken your comments into consideration and reduced 2 self-references to ensure that the introduction provides relevant and concise background information.

Comments and Suggestions 7#: The article lacks a detailed explanation of the molecular mechanisms underlying the observed effects of probiotics on meat quality and lipid metabolism. The article lacks a discussion of the potential risks and drawbacks of using probiotics as feed additives. It is important to consider the potential negative effects of probiotics, such as the development of antibiotic resistance and the potential for unintended consequences on the gut microbiota of animals.

Response: Please forgive me for not mentioning these two parts in the article. Although the inspiration for this study comes from Bacillus LFB112 to improve the meat quality of broilers and to increase the content of unsaturated fatty acids in muscle tissue, the purpose of this experiment is to conduct in vitro experiments to study the regulation of fatty acid metabolism on Bacillus LFB112.

 A detailed explanation of the molecular mechanisms underlying the effects of probiotics on meat quality and lipid metabolism will appear in another paper from our laboratory, which has not yet been published, If you are interested in this section, you can follow the subsequent related articles in our laboratory.

The discussion on the potential risks and drawbacks of using probiotics as feed additives have been mentioned in previous animal experiments published in our laboratory, as this article mainly tells the story of the fatty acid metabolism of Bacillus LFB112 in vitro culture, so there is no discussion on the potential risks of using probiotics as feed additives. If you are interested in this section, you can follow the articles about animal experiments in our laboratory.

Comments and Suggestions 8#: Conclusions must be revised to reflect the findings

Response: The purpose and conclusion of the experiment have been revised in the revised manuscript, hoping to make the article's ideas clearer, Thank you for your feedback on this section.

Comments and Suggestions 9#: References are not according to the journal guidelines

Response: Thank you very much for your feedback on this part of the References. In the process of revising the manuscript, I standardized the writing of bacterial names in the references and simplified the citation of published articles in our laboratory, hoping to meet the requirements of the journal for references

Comments and Suggestions 10#: It is unclear how the authors confirmed the fatty acid metabolism of Bacillus LFB112.

Response: In this study, we investigated the fatty acid metabolism of Bacillus LFB112 through a combination of changes in fatty acid content, genomic annotation, and enzyme and metabolite analyses. Initially, we compared the changes in fatty acid content in the supernatant and pellet of Bacillus LFB112 cultured on BPY medium and BPY+soybean oil medium, allowing for visual confirmation of the fatty acid metabolism in Bacillus LFB112. Subsequently, we performed KEGG annotation of the entire genome data of Bacillus LFB112, revealing that its fatty acid synthesis and metabolism are predominantly regulated by the type II fatty acid synthesis pathway. Moreover, we conducted quantitative detection of enzyme mRNA level and intermediate metabolites associated with the type II fatty acid synthesis system in Bacillus LFB112, providing molecular-level evidence of the changes in fatty acid metabolism in Bacillus LFB112. These findings collectively demonstrate that Bacillus LFB112 possesses the capability to metabolize fatty acids.

Comments and Suggestions 11#: what specific methods they used to measure the expression levels of genes related to fatty acid metabolism

Response: In the present study, a comprehensive analysis of the entire genome data of Bacillus LFB112 was conducted using KEGG annotation, revealing a significant abundance of genes related to fatty acid metabolism, with a particular enrichment of genes associated with the type II fatty acid synthesis pathway. To further elucidate the transcriptional activity of key enzymes involved in this pathway, primers were designed for target enzymes, and total RNA was extracted from Bacillus LFB112 cells. Subsequently, qPCR was employed, following reverse transcription, to quantitatively assess the mRNA levels of the relevant enzymes.

I hope this revised version provides a clearer and more detailed explanation of the study. If you have any further suggestions or questions, please feel free to let me know.

Sincerely,

Cheng Qiang
